# Targeted Delivery of Nanoparticulate Cytochrome C into Glioma Cells Through the Proton-Coupled Folate Transporter

**DOI:** 10.3390/biom9040154

**Published:** 2019-04-18

**Authors:** Yuriy V. Kucheryavykh, Josue Davila, Jescelica Ortiz-Rivera, Mikhael Inyushin, Luis Almodovar, Miguel Mayol, Moraima Morales-Cruz, Alejandra Cruz-Montañez, Vanessa Barcelo-Bovea, Kai Griebenow, Lilia Y. Kucheryavykh

**Affiliations:** 1Department of Biochemistry, Universidad Central del Caribe, School of Medicine, Bayamon, PR 00956, USA; yuriy.kucheryavykh@uccaribe.edu (Y.V.K.); 416jortiz@uccaribe.edu (J.O.-R.); 2Universidad Central del Caribe, School of Medicine, Bayamon, PR 00956, USA; jdavilaro41@gmail.com; 3Department of Physiology, Universidad Central del Caribe, School of Medicine, Bayamon, PR 00956, USA; mikhail.inyushin@uccaribe.edu; 4HIMA San Pablo Hospital, Caguas, PR 00726, USA; lj_almodovar@yahoo.com; 5Department of Neurosurgery, University of Puerto Rico, Medical Sciences Campus, Rio Piedras, PR 00935, USA; miguel.mayol@upr.edu; 6Department of Chemistry, University of Puerto Rico, Río Piedras Campus, San Juan, PR 00931, USA; moraima84@gmail.com (M.M.-C.); ac_montanez@hotmail.com (A.C.-M.); vanessabarcelo1@gmail.com (V.B.-B.); kai.griebenow@gmail.com (K.G.)

**Keywords:** glioma, proton-coupled folate transporter, folate receptor 1

## Abstract

In this study, we identified the proton-coupled folate transporter (PCFT) as a route for targeted delivery of drugs to some gliomas. Using the techniques of confocal imaging, quantitative reverse transcription-polymerase chain reaction (qRT-PCR), and small interfering (siRNA) knockdown against the PCFT, we demonstrated that Gl261 and A172 glioma cells, but not U87 and primary cultured astrocytes, express the PCFT, which provides selective internalization of folic acid (FA)-conjugated cytochrome c-containing nanoparticles (FA-Cyt c NPs), followed by cell death. The FA-Cyt c NPs (100 µg/mL), had no cytotoxic effects in astrocytes but caused death in glioma cells, according to their level of expression of PCFT. Whole-cell patch clamp recording revealed FA-induced membrane currents in FA-Cyt c NPs-sensitive gliomas, that were reduced by siRNA PCFT knockdown in a similar manner as by application of FA-Cyt c NPs, indicating that the PCFT is a route for internalization of FA-conjugated NPs in these glioma cells. Analysis of human glioblastoma specimens revealed that at least 25% of glioblastomas express elevated level of either PCFT or folate receptor (FOLR1). We conclude that the PCFT provides a mechanism for targeted delivery of drugs to some gliomas as a starting point for the development of efficient methods for treating gliomas with high expression of PCFT and/or FOLR1.

## 1. Introduction

Glioma is one of the most malignant forms of brain cancer. Current chemotherapeutic treatments typically lack specificity against cancer cells and are associated with significant toxicity on healthy tissues, affecting the overall health of the patient. Therapeutic approaches based on targeted nanoparticulate drug delivery open a new avenue for cancer treatment and allow selective binding to cancer cells, with minor effects on healthy tissues [1,2,3]. Recently, we developed a system for specific and direct introduction of apoptosis-inducing drugs into cancer cells [4,5,6]. The system is based on conjugation of nano-particulate cytochrome c (Cyt c NPs) to ligands, which are specific for receptors in tumors, in order to enhance selective cellular uptake and initiate apoptosis. This designed system demonstrated an increased potential therapeutic benefit, based on the ability to deliver high drug concentrations to the tumor site without the need for excessive systemic exposure [7]. 

The therapeutic potential of nano-encapsulated cytochrome c and granzyme B has previously been demonstrated for lung carcinoma and hepatoma cell lines [8,9,10], and recently, we demonstrated in mice, both in vivo and in vitro, the efficacy of Cyt c NPs for the treatment of gliomas [7]. However, the optimum vehicle for specific drug delivery into brain tumors is still in question. Analysis of glioblastoma samples from adult human patients has indicated that these tumors overexpress several receptors that distinguish them from the surrounding brain parenchyma. Urokinase-type plasminogen activator, transferrin, EphA2, and IL-13 are largely absent in the normal brain, but overexpressed in gliomas [11,12,13,14,15,16,17] and can mediate endocytosis of corresponding ligand-conjugated anticancer nanomedicines. Several recombinant proteins consisting of toxins fused to ligands of known protein receptors have been developed to deliver toxic agents to glioma cells. However, multiple problems associated with low specificity for tumors and dose-related neurological side effects have been identified [11]. Selection of specific ligands, providing effective and selective delivery of drug to glioma tumors, is still under investigation. 

Folic acid (FA) has been extensively studied as a ligand–drug conjugate for delivery into tumor cells [18,19]. It is a basic component of cell metabolism and DNA synthesis and repair and is essential for both normal and tumor cells [20]. However, as an essential molecule in the DNA synthesis pathway, FA is critical for cancer cell duplication. Compared with normal cells, some tumors overexpress folate receptors, which have highly efficient recycling pathways and can mediate endocytosis of corresponding ligand-conjugated anticancer nanomedicines [11,18,21]. Using a HeLa cell model, a folate receptor overexpressing carcinoma cell line [22], we demonstrated that the FA targeting moiety significantly enhances the cytotoxic effect of Cyt c-based NP compared to NPs without ligand [7]. This was also confirmed for a number of other nano-constructs [23,24,25,26,27]. Unfortunately, FA–drug conjugates have not been well studied as drug delivery vehicles for adult gliomas, since folate receptors are limited in these tumors [19,28]. However, we recently demonstrated the efficacy of FA-decorated Cyt c NPs in a C57Bl/6 mouse glioma model [7]. In the current study, we show the successful internalization of fluorescently conjugated FA-Cyt c NPs in some of glioma cell lines. In this work, we evaluate the possibility of a folate-receptor-mediated endocytosis mechanism as the cellular internalization machinery of our FA-decorated Cyt c NPs. Despite low expression of FA receptors in cultured glioma cells, we identified the proton-coupled folate transporter (PCFT), which provides the major FA-pumping mechanism in some of glioma cell cultures. By contrast, low expression of the PCFT was detected in normal astrocytes. The PCFT (encoded by the SLC46A1 gene) is a proton–FA symporter, with an acidic pH optimum that couples the flow of protons down an electrochemical concentration gradient to the uptake of folates into cells and has been shown to mediate endocytosis of FA-conjugated nano-drugs [29]. 

We have demonstrated the presence and functional activity of the PCFT in some cultured glioma cells and provided evidence that this transporter is involved in the internalization of FA-Cyt c NPs into PCFT expressing gliomas. Our findings support the hypothesis that FA is a suitable ligand candidate in the development of drugs for targeted treatment of certain gliomas, expressing high levels of folate receptors and/or PCFT, and could provide successful and specific internalization of nanoparticulate drugs.

## 2. Results

### 2.1. Cultured Glioma Cells Specifically Take Up Folic Acid-Conjugated Cytochrome c-Containing NanoFA-Cyt c NP Constructs

The specificity of internalization of the designed FA-Cyt c NP constructs by cultured glioma cells was tested by using FA-Cyt c NPs modified with fluorescein isothiocyanate (FITC) via the amine group. Human A172, U87, mouse Gl261 glioma cells, and mouse cultured primary astrocytes were treated with FITC–labeled FA-Cyt c NPs, added to the culture medium for a final concentration of 100 µg/mL (100 mg/mL stock solution was prepared by dilution of designed NPs in phosphate buffer saline (PBS)). Cells were cultured in neutral (pH = 7) and acidic (pH = 6) culture medium, optimal for PCFT activity. Confocal imaging of live cells revealed the robust accumulation of FITC fluorescence in GL261 and A172 cells after six hours of incubation with FITC–labeled FA-Cyt c NPs (Figure 1A). Astrocytic cells did not show any sign of uptake of NPs at this time point. Additionally, A172 and GL261 cells internalized NPs better in the acidic environment compared to the neutral pH (Figure 1B). U87 cells showed low uptake in both pH = 6 and pH = 7. An acidic pH of the cell culture medium did not significantly affect cell viability during the experiment (Appendix A).

### 2.2. Folic Acid-Conjugated Cytochrome c-Containing Nanoconstructs Cause Cell Death in Glioma Cells but Not in Astrocytes

A live/dead cell assay was performed for Gl261, A172, U87 glioma cells, and for mouse primary cultured astrocytes. To evaluate the efficacy of application of FA-Cyt c NP constructs (folate-poly(ethylene glycol)-poly(lactic-co-glycolic acid) conjugate Cyt c-based NPs (FA-PEG-PLGA-Cyt c) in the glioma model, cells were seeded in petri dishes, and FA-Cyt c NPs (100 µg/mL) were added to the culture medium and incubated for 24 h. Phosphate buffer saline without NPs was added to control cells in the same volume. 100 µg/mL of FA-PEG-PLGA polymer not containing Cyt c were used as an additional control in order to monitor the cytotoxicity of the delivery system. Primary cultured astrocytes received the same treatments, with the purpose of assessing the specificity of drug constructs designed for glioma cells. 

The results demonstrated 40% cell death for Gl261 cells and 30% cell death for A172, but not for astrocytes and U87 cells, in a 24-h treatment with FA-Cyt c NPs (Figure 2). The FA-PEG-PLGA exhibited no cytotoxic effects in the entire cell cultures investigated. Extended treatment with FA-Cyt c NPs for up to five days did not show any cytotoxic effect on astrocytes (Appendix A), confirming the specificity of the FA-conjugated nanoconstructs for GL261 and A172 cells.

### 2.3. Glioma Cells Specifically Take Up Folic Acid-Conjugated Cytochrome c-Containing Nanoconstructs Through the Proton-Coupled Folate Transporter Mechanism

The mechanism of internalization of FA-conjugated nanoconstructs was investigated with electrophysiological recordings of membrane FA currents in glioma cells. The whole-cell voltage clamp (held at −60 mV) was elicited by application of FA (100 µM) in the extracellular solution. The currents induced by FA were registered at pH 6.0 (Figure 3A) with the highest magnitude detected for Gl261 cells and the lowest for U87 cells. These currents were blocked by the application of FA-Cyt c NPs in a concentration-dependent manner, indicating the competitive nature of these substrates (Figure 3B,C). These results confirm that binding and internalization of FA-Cyt c NP constructs occurs by means of an FA-specific carrier in the plasma membrane of glioma cells. The maximum currents induced by FA at the acidic pH condition allow us to propose the PCFT as the most plausible candidate for the FA carrier, as acidic pH has been demonstrated to be favorable for PCFT activity [30]. Small interfering RNA (siRNA) knockdown of the PCFT resulted in the reduction of currents induced by application of FA in GL261 cells (Figure 3D), indicating that PCFT activity is the key mechanism of FA transport in GL261 cells (efficacy of the siRNA knock-down is shown in Appendix A).

Immunocytochemical experiments revealed the robust expression of the PCFT in GL261 and A172 cells, but not in astrocytes (Figure 4A). Expression of PCFT in A172 cells were lower compared to GL261, but significantly higher compared to U87 cells. U87 cells showed the lowest level of PCFT expression in investigated glioma cell lines. These results were confirmed by quantitative reverse transcription-polymerase chain reaction (RT-PCR), indicating a low PCFT gene expression in astrocytes, limited PCFT gene expression in U87 human glioma cells, higher expression in A172, and strong PCFT expression in GL261 cells (Figure 4B). Some limited expression of PCFT was detected in a whole cortex tissue in mice, which is, however, significantly lower compared to the mouse GL261 glioma cells. No significant expression of folate receptor 1 (FOLR1) was detected in glioma cells. Kidney and gut tissues from mice were used as a positive control for PCFT gene expression, as these tissues are shown to have a high level of PCFT and FOLR1 expression [30,31]. 

Small interfering RNA knockdown of the PCFT prevented internalization of FA-Cyt c NPs by GL261 cells and reduced cell death in response to FA-Cyt c NPs treatment. Figure 5A shows the confocal image of live GL261 cells after six hours of incubation with FITC-conjugated FA-Cyt c NPs with and without PCFT knockdown. Non-transfected and mock-transfected cells demonstrated accumulation of fluorescence in cytoplasmic structures, while anti-PCFT-transfected cells did not accumulate fluorescence in the cytoplasm, and fluorescent particles were observed in the extracellular medium. The influence of siRNA knockdown of the PCFT on GL261 cell viability is presented in Appendix A. Data indicate that PCFT knockdown did not result in increased cell death but reduced cell viability, indicating the reduction of cell growth. Additionally, FA-Cyt c NPs demonstrated reduced cytotoxicity in anti-PCFT-transfected cells compared to MOCK transfected ones (Figure 5B), indicating that PCFT is the major uptake route for FA-Cyt c NPs targeting GL261 cells. 

### 2.4. Overexpression of Proton-Coupled Folate Transporter and Folate Receptor 1 Were Found in Some Human Glioblastoma Specimens

Nine human glioblastoma specimens were investigated in order to evaluate the level of gene expression of PCFT (SLC46A1) and FOLR1 (data presented in Table 1). Tissue samples were homogenized using collagenase/hyaluronidase and glioma cells were purified from the specimens with use of Percoll gradients, in order to remove blood and endothelium cells that might express PCFT and/or FOLR1. Quantitative RT-PCR analysis revealed the low level of expression of PCFT in two of nine specimens (patients 3 and 7) and elevated expression of PCFT in five specimens (1, 4, 6, 8 and 9), with one of them elevated drastically (9). Analysis of FOLR1 showed strong up-regulation of FOLR1 expression in four of nine patients (1, 3, 6 and 9), with low expression in five of nine patients. Notably, six of nine patients had higher level of expression of PCFT compared to FOLR1, with three patients showing significant overexpression of FOLR1 over PCFT. 

These data correlate with the statistical information, obtained from The Cancer Genome Atlas (TCGA) glioblastoma samples project. The gain of copy number variations (CNV) for PCFT (SLC46A1), compared to the normal brain tissue, was detected in 25% of glioblastoma cases, with CNV loss in 68.75% (data obtained from 16 cases). Copy number variations gain for FOLR1 was detected in 16.67% of cases, with CNV loss in 58.33% (data obtained from 19 cases). In comparison to glioblastoma, the gain of CNV for PCFT was not detected in low grade gliomas, while CNV loss was found in 77.78% (data obtained from nine cases). On the contrary, CNV gain for FOLR1 was detected in 57.14% of low-grade gliomas, with 28.57% of CNV loss (data obtained from seven cases). These allow the conclusion that more than 25% of glioblastoma tumors express elevated levels of either one or both of PCFT and FOLR1, respectively, and more than 50% of low grade gliomas overexpress FOLR1. This suggests that FA carriers can be used for the development of targeted drug delivery mechanisms. 

### 2.5. Local in-TumorIintroduction of Folic Acid-Conjugated Cytochrome c-Containing Nanoparticles s Increases the Survival Rates in GL261/C57Bl/6 Mouse Glioma Model

The efficacy of the FA-Cyt c NPs system was tested in a GL261/C57Bl/6 mouse glioma model. Previously, we reported that local in-tumor introduction of FA-Cyt c NPs led to the reduction of tumor size and the presence of signs of massive apoptosis in tumor tissue, but not in healthy brain parenchyma in GL261/C57Bl/6 model [7]. Here, survival analysis was performed (Figure 6). Two groups of glioma-bearing mice (six animals per group) received either FA-Cyt c NPs (100 mg/mL diluted in PBS) or vehicle (PBS). The drug was infused directly into the tumor, beginning on the 7th day after tumor implantation with the use of a mini-osmotic pump, installed subcutaneously, at 1 μL/h, over 14 days. The median survival in the group receiving FA-Cyt c NPs was 23.5 days after glioma implantation vs. 20 days in the control group. This difference of 3.5 days represents a 17.5% increase in survival in response to the FA-Cyt c NPs treatment. 

## 3. Discussion

In this study, we demonstrated that the PCFT mechanism can be exploited for FA-directed delivery of nanodrugs to certain gliomas. Cancer cells are highly dependent on FA for DNA synthesis and replication, and the overexpression of FA receptors has been demonstrated for many cancers compared with noncancerous cells [32,33]. This differential expression of FA receptors in cancer and somatic cells makes FA an excellent ligand carrier for targeted anticancer treatment, leading to the robust intracellular accumulation of FA-tagged nanodrugs in cancer cells and allowing reduction of drug exposure in healthy tissue. Significant anticancer effects of FA-conjugated drugs have been shown in ovarian, prostate, and other cancers [34,35,36]. Conjugation with FA enables specific targeting of cancer cells and prevents undesirable side effects of the treatment, making it an attractive approach to drug development. 

However, FA has not been used for targeted drug delivery to glioma tumors, because FA receptors were not found in many of these cancers. Nevertheless, we recently demonstrated the robust internalization of FA-decorated Cyt c-based NPs by tumors, but not by healthy brain tissue in a tumor-bearing GL261/C57BL/6 mouse glioma model [7]. The uptake of FA-Cyt c NPs was accompanied by signs of massive apoptosis in implanted GL261 tumors and by reduction of tumor growth. In the present study, we confirmed in vitro the specificity of FA-Cyt c NP constructs in targeting some gliomas compared with healthy cultured astrocytes. We demonstrated that some of the investigated glioma cell lines (GL261 and A172), but not astrocytes and U87 cells, strongly take up FA-Cyt c NPs followed by extensive cell death. The FA-Cyt c NPs up-take was correlated with the level of expression of PCFT (Figure 4). Observation of astrocytes over five days did not reveal any long-term effects of FA-Cyt c NPs on astrocyte viability, indicating a specific cytotoxic effect on glioma cells expressing PCFT. The combination of electrophysiological and confocal live cell imaging approaches revealed a PCFT-dependent FA mechanism and FA-Cyt c NP internalization by GL261 and A172 cells (Figure 3; Figure 5). The siRNA knockdown of the PCFT in GL261 cells resulted in significant reduction of FA-induced electrical currents at pH 6.0 (Figure 3D) and also in reduction of FITC-conjugated FA-Cyt c NP uptake by glioma cells (Figure 5). Analysis of glioblastoma human specimens revealed that a significant amount of tumors express elevated levels of PCFT and/or FOLR1. These results indicate that PCFT activity, together with FOLR1, is the one of the key mechanisms of FA pumping and FA-Cyt c NP uptake in some gliomas. Together with results obtained by quantitative gene expression analysis and immunofluorescence demonstrating the robust expression of the PCFT in GL261 and A172 cells, but low expression of PCFT in astrocytes and healthy brain parenchyma, these results indicate that the functional activity of the PCFT in some glioma cells represents the route of targeted and specific administration of FA-conjugated drugs into glioma cells. 

The abundant expression of the PCFT and its associated FA transport activity at low pH has been reported for some human tumors in lung, breast, prostate, and ovarian cancers [37,38]. Recently, expression of the PCFT in resected human glioma specimens and in human glioma cell lines has also been described [39]. The tumor microenvironment is characterized by acidic pH levels due to rapid metabolism and cell division [30,40,41] and therefore provides the optimum conditions for PCFT activity. The additional level of selectivity of a PCFT-dependent mechanism of drug delivery to tumor tissue is based on the fact that pH in normal healthy tissues is maintained within the neutral range, and the PCFT expressed in these tissues is not expected to be involved in active pumping of FA and FA-conjugated drugs. Despite the fact that expression of the PCFT was observed in several tissues, in tissues where the extracellular milieu is at neutral pH, such as at the apical brush-border membrane of the duodenum and proximal jejunum, the sinusoidal membrane of the liver, the apical brush-border membrane of the kidney, the basolateral membrane of the choroid plexus, the retinal pigment epithelium, and the placenta [30,42,43,44,45], the role of the PCFT is based on a different mechanism. For these tissues the PCFT has been shown to be involved in FA receptor-mediated endocytosis [44], in which the PCFT serves as an endosomal FA exporter providing the export of folates from acidified endosomes. The ubiquitous expression of the PCFT in mammalian tissues may be related to this function, but not to FA pumping from the extracellular milieu. 

The mechanism of internalization of FA-conjugated NPs through the PCFT has not yet been studied and requires additional investigation. The size of FA-Cyt c NPs does not allow efficient pumping of the particle to the cellular cytosol, resulting in binding to and blocking of the transporter. Therefore, PCFT-dependent endocytosis is proposed. Several classes of receptors and transporters that promote endocytosis have been characterized, including the FA receptor, the glucose transporter, and the glutamic acid transporter [44,45,46,47]. Additionally, endocytosis serves as a way to reduce transporters from the plasma membrane for their downregulation or as a mechanism of non-functional or disabled transporters. Prior to being endocytosed, transporters are subjected to ubiquitination, which acts as an endocytosis signal [48,49]. Both mechanisms of internalization of FA-Cyt c NPs bound to the PCFT are possible.

The observed effect on glioma cell death (40% in GL261 and 30% in A172 cell lines) was less prominent compared to the effect on HeLa cells (80% cell death), that was observed in our previous study [7]. Taking into account the high level of Cyt c PN internalization in these glioma cell lines, the increased activation of anti-apoptotic signaling mechanisms can be proposed in gliomas. It has been shown that Akt signaling is highly activated in many gliomas, mediated by the overexpression of receptor tyrosine kinases, i.e., epidermal growth factor receptor (EGFR) and platelet derived growth factor receptor (PDGFR) [50,51,52,53,54]. Activation of Akt and STAT signaling were shown to lead to the apoptosis inhibition, increased survival and treatment resistance, including resistance to chemo- and radio-therapy [55,56]. To bypass the apoptosis inhibition and to enhance sensitivity to Cyt c in gliomas the combinatorial treatment targeting EGFR or PDGFR pathways could be proposed.

The developed FA-Cyt c NPs system can potentially be used to deliver other protein-based therapeutics into tumor cells with high expression of FOLR1 and/or PCFT. Limitations include tumors with high interstitial pressure and very dense tumors (e.g., pancreatic cancer) because of delivery problems in general. Small therapeutic molecules can be incorporated in the system through the co-nanoprecipitation with the protein; however, it would be limited to hydrophilic drugs that can be co-precipitated with Cyt c. A further limitation might be the tumor insensitivity to Cyt c. 

In this study, we used in-tumor delivery of FA-Cyt c NPs in order to bypass potential problems related to blood-brain barrier permeability for nano-constructs. The intranasal administration can be considered as an alternative route for the delivery of nano-particulate drugs to brain tumors. Olfactory and vomeronasal routes for the delivery of plasmids and big macromolecules may provide direct access to certain regions of the brain bypassing the blood-brain barrier and blood-cerebrospinal fluid barrier [57]. The intranasal route was shown to be effective for brain delivery of lipid nanoparticles and liposomes [58]. Studies performed using a rat animal model demonstrated that 48.15% of folic acid containing niosomes (nanoparticles in the size range of 3.05–5.625 µm) was absorbed through nasal cavities during six hours after the drug administration [59]. More detailed studies are needed for the evaluation of the efficacy of intranasal administration of protein-based nano-particulate drugs to brain tumors and possible off-target effects. In conclusion, this study revealed the specific uptake and cytotoxic effect of FA-conjugated Cyt c NPs in glioma cells with high level of expression of PCFT, but not in healthy astrocytes. Efficacy of Cyt c NPs take up in glioma cells correlated with the level of expression of PCFT, providing the mechanism of internalization of FA-Cyt c NPs. The unique biological feature of some glioma tumors, which includes the increased expression of the PCFT and an acidic microenvironment, provides favorable conditions for PCFT activity and establishes the feasibility of selective delivery of FA-conjugated drugs to glioma tumors via the PCFT. However, it has been shown that the PCFT is involved in the transport of folates into the central nervous system [44,60,61], and is required for FA transport across the blood–choroid plexus–cerebrospinal fluid barrier. For this reason, more detailed studies aimed at the evaluation of possible effects on healthy brain tissues are needed. The conventional method of local in-tumor drug administration can be proposed in order to avoid unwanted effects on possibly vulnerable brain structures as well as to avoid low drug permeability of the blood–brain barrier.

## 4. Materials and Methods

All experimental procedures were carried out in accordance with the protocols approved by the Institutional Animal Care and Use Committee (IACUC, protocol # 029-2017-25-01-PHA-IBC) and with use of broad consent approved by the Institutional Review Board (IRB) Human Research Subject Protection Office (protocol #2012-12B).

### 4.1. Cell Culture

The GL261 glioma cell line was obtained from the NCI (National Cancer Institute, Frederick, MD, USA). The A172 and U87 human glioma cell line was obtained from American Type Culture Collection (ATCC) (Manassas, VA, USA). Primary astrocyte cultures were prepared from the neocortex of 1–2-day-old C57BL/6 mice as previously described [62]. Briefly, brains were removed after decapitation and the meninges stripped away to minimize fibroblast contamination. The forebrain cortices were collected and dissociated using a Stomacher blender. The cell suspension was then allowed to filter by gravity through a #60 sieve and then through a #100 sieve. After centrifugation, the cells were suspended in Dulbecco’s modified Eagle’s medium (DMEM, Sigma-Aldrich, St. Louis, MO, USA) containing 25 mM glucose, 2 mM glutamine, 1 mM pyruvate, and 10% fetal calf serum, and plated on uncoated 75-cm^2^ flasks at a density of 300,000 cells/cm^2^. At confluence (after about 12–14 days), the mixed glial cultures were treated with 50 mM leucine methylester (pH 7.4) for 60 min to kill microglia. Cultures were then allowed to recover for at least one day in growth medium prior to reseeding. Astrocytes were dissociated by trypsinization and reseeded onto appropriate plates for experiments. 

All cells were cultured in DMEM supplemented with 10% fetal calf serum, 0.2 mM glutamine, and antibiotics (50 U/mL penicillin, 50 µg/mL streptomycin) and maintained in a humidified atmosphere of CO_2_/air (5%/95%) at 37 °C. The medium was exchanged with fresh culture medium about every 2–3 days.

### 4.2. Synthesis of FA-PEG-PLGA-S-S-Cyt c NPs

Cyt c nanoparticles were obtained using the methodology recently described by us [7]. Ten mg/mL of Cyt c in the presence of the excipient methyl-β-cyclodextrin in nanopure water was solvent-precipitated by adding acetonitrile at a 1:4 volume ratio. These nanoparticles are made stable in aqueous solution by coating with a thiol-end amphiphilic polymer folate-poly(ethylene glycol)-poly(lactic-co-glycolic acid)-thiol (FA-PEG-PLGA-SH) conjugate synthesized by us [7]. Briefly, following the Cyt c nanoprecipitation, the Succinimidyl-3-(2-pyridyldithio)propionate (SPDP, Proteochem, Denver, CO, USA) linker (0.8 mg, 1:3 Cyt c-to-linker molar ratio) was added directly to the resulting suspension to modify the nanoparticle surface with the linker. After 30 min, 30 mg of FA-PEG-PLGA-SH (0.0009 mmol) was dissolved in 10 mL of acetonitrile and added to the mixture to form a thiol bond with the Cyt c attached linker. The mixture was allowed to react at RT for 18 h. To stop the reaction and to remove unreacted reagents, the nanoparticles were washed by multiple cycles of resuspension/centrifugation (8000 rpm, 15 min). Throughout this article we adopt the nomenclature of FA-Cyt c NPs to name the resulting Cyt c-based NPs coated with FA-PEG-PLGA.

### 4.3. In Vitro Live/Dead Assay

Live/dead assays were performed with the live/dead viability/cytotoxicity Kit (Invitrogen, #L3224, Carlsbad, CA, USA), based on calcein and ethidium homodimer-1 staining of live and dead cells, respectively. Cells were seeded in 24-well dishes and incubated with Cyt c-NPs decorated with FA (FA-PLGA-PEG-Cyt c NPs), 100 µg/mL final concentration, for 24 h (a stock solution of 100 mg/mL was prepared in PBS). Control cells received treatment with PBS or FA-PEG-PLGA polymer not containing Cyt c in the same volumes and concentrations. After 24 h, the cells were stained with calcein (4 µM) and ethidium homodimer-1 (2 µM) diluted in C-PBS for 30 min at room temperature. Images of the cells were captured immediately after the staining using an inverted microscope (Olympus IX71, Tokyo, Japan), Qcolor 3 camera (Olympus), and Q-capture Pro software (Olympus). To estimate the relative number of dead cells, ten images were taken for each well. Live and dead cells were counted in each image and the relative number of dead cells was calculated as a percentage of the total number of cells. 

### 4.4. FA-NP Cyt C Uptake Assay

Cells were seeded on coverslips (#1.5; Warner Instruments, Hamden, CT, USA) and incubated with FITC-conjugated FA-Cyt c NPs (100 µg/mL) for six hours. Coverslips with the cells were then transferred on microscope slides and immediately visualized using an Olympus Fluoview FV1000 confocal microscope (Olympus, Tokyo, Japan) with a 40× oil-immersion objective. Fluorescent cells that had taken up FITC-conjugated FA-Cyt c NPs were identified using a FITC excitation–emission filter set (absorption maximum at 494 nm and emission maximum of 521 nm). The fluorescent images were processed using free Image J imaging software [63].

### 4.5. Electrophysiological Recording from Single Cells

Coverslips with cultured cells were transferred to a recording chamber (RC-27L Warner Instr. Corp., Hamden, CT, USA) adapted to the stage of an Olympus upright microscope with infrared and fluorescence attachments. Cells were visualized using a Nomarski optical infrared attachment equipped with differential interference contrast (DIC, BX51WI Olympus) and a DP30BW digital camera with DP Controller software (Olympus). The piezoelectric micromanipulator (MX7500 with MC-1000 drive, Siskiyou, Inc., Grants Pass, OR, USA) was used for precise positioning of micropipettes. Membrane current was recorded from GL261 cells using whole-cell voltage clamp. The extracellular perfusion solution (ECS) contained (in mM): NaCl, 140; CaCl_2_, 2.5; MgCl_2_, 2; 4-(2-hydroxyethyl)-1-piperazineethanesulfonic acid (HEPES), 10; and KCl, 3 (osmolarity was kept constant at 308 mOsmol/L). Extracellular pH was adjusted to pH 6.0. FA (Sigma Chemical Co., St. Louis, MO, USA) was used to initiate intracellular current. HEKA amplifiers (EPC-10, 3 channels, HEKA Elektronik, Lambrecht/Pfalz, Germany) were used to acquire and store the data obtained from cells. Data were analyzed using Origin 9.1 software (OriginLab, Northampton, MA, USA).

Patch pipettes were fabricated from borosilicate glass capillaries OD 1.5/ID 0.86 mm (#1B150F-4, World Precision Instr., Sarasota, FL, USA) using a P-1000 puller (Sutter Instr. Co., Novato, CA, USA). The patch pipettes were filled with a pipette solution containing (in mM): KCl, 140; HEPES-KOH, 10; Ethylene Glycol Tetraacetic Acid (EGTA), 2; CaCl_2_, 0.2; MgCl_2_, 1; pH 7.2. Resistance of the pipettes was in the range of 5–8 MΩ.

### 4.6. Immunocytochemistry

Immunostaining was performed using the protocol previously established in our lab [64]. Cells seeded on coverslips were fixed with 4% Paraformaldehyde (PFA, Sigma-Aldrich, St. Louis, MO, USA). The cells were then blocked with 5% normal goat serum/5% normal horse serum (Vector lab., Burlingame, CA, USA) in PBS containing 0.3% Triton X-100 and 0.05% phenylhydrazine for 30 min and incubated with rabbit polyclonal anti-SLC46A1 or anti-FOLR1 antibody (# TA334591, #TA324367, OriGene Technologies Inc., Rockville, MD; USA), diluted 1:500 in PBS-TAT (0.3% TritonX-100, 5% normal goat/5% normal horse serum, 1% sodium azide, 0.01% thimerosal) overnight at 4 °C. The cells were incubated with corresponding secondary antibodies (FITC anti-rabbit IgG Vector Lab., Burlingame, CA, USA) overnight and visualized using an Olympus Fluoview FV1000 confocal microscope with 40× oil-immersion objective.

### 4.7. RNA Interference by Small Double-Stranded RNAs

GL261 cells were transfected with siRNAs targeting SLC46A1 (Qiagen, cat. #SI00965559, GmbH, Germany) using HiPerfect Transfection reagent according to the manufacturer’s instructions and as we previously described [63]. One hundred microliters of serum-free medium containing 2 µL of 20 nM SLC46A1 siRNA and 20 µL of HiPerfect reagent were prepared and incubated for 30 min at room temperature. This complex was then added to 5 × 10^4^ glioma cells containing 1.9 mL of cell culture medium in a drop-wise fashion and the plate gently swirled to evenly distribute the transfection complex. In addition, mock transfections, in which 100 µL of serum-free medium containing 20 µL of HiPerfect reagent without siRNA was added to the cells, were performed and used as a control. A time point of three days after transfection was used. The efficiency of SLC46A1 knockdown was determined using qRT-PCR. 

### 4.8. Quantitative RT-PCR Analysis

Expression of SLC46A1 and FOLR1 was analyzed using qRT-PCR. RNA was extracted from cell pellets using RNeasy Plus Mini Kit (Qiagene GmbH, Hilden, Germany) following the manufacturer’s protocol. RNA quality and concentration was quantified spectrophotometrically with the NanoDrop 1000 spectrophotometer (Thermo Scientific, Waltham, MA, USA). Complementary DNA (cDNA) was reverse-transcribed from 1 µg of total RNA using the iScript cDNA synthesis kit (Bio-Rad, Hercules, CA, USA). An asymmetrical cyanine dye SYBR green qRT-PCR gene expression assays were performed using 50 nM Hs_SLC46A1 and Hs_FOLR1 primers (#QT00012488, #QT00067515, Qiagene, GmbH). Amplification was carried out in a Bio-Rad CFX96 Touch real-time PCR detection system (Bio-Rad). 

The gene expression level was defined as the threshold cycle number (CT). Mean fold changes in expression of the target genes were calculated using the comparative CT method (RU, 2–ΔΔCt). All data were controlled for the quantity of RNA input by GAPDH forward 5′-CTGGGCTACACTGAGCACC-3′; GAPDH reverse 5′-AAGTGGTCGTTGAGGGCAATG-3′ serving as the endogenous control and for normalization (results were normalized with the control column).

### 4.9. Glioblastomatissue Samples Acquisition and Glioblastoma Cells Purification

Glioblastoma specimens were obtained from HIMA San Pablo Hospital (Caguas, PR, USA) and from the University of Puerto Rico Medical Center (Rio Piedras, PR, USA). Tissue samples were dissected to 1–2 mm pieces with a razor blade, and homogenized using collagenase (300 U/mL) /hyaluronidase (100 U/mL) (StemCell Technologies, Tukwila, WA, USA), diluted in DMEM. Glioma cells were purified from the homogenized tissue using Percoll (Sigma-Aldrich, St. Louis, MO, USA) gradients of 30% and 70%. Following this procedure, the glioma fraction was collected from the top of 70% Percoll level. Glioma fraction was used for further analysis.

### 4.10. Patient Datasets

Copy number variation statistical data (Affymetrix SNP 6.0 platform) of 27 glioblastoma cases from the Cancer Genome Atlas project (TCGA-GBM) were obtained from the NIH GDC Data Portal (https://portal.gdc.cancer.gov). 

### 4.11. In Vivo Glioma Implantation and Survival Analysis

All surgery was performed under isoflurane anesthesia, and all efforts were made to minimize suffering. The GL261 glioma cells were implanted into the right cerebral hemisphere of 12–16-week-old C57BL/6 mice. Implantation was performed according the protocol that we described earlier [7,64]. Mice were anesthetized with isoflurane and a midline incision was made on the scalp. At stereotaxic coordinates of bregma, 2 mm lateral, 1 mm caudal and 3 mm ventral a small burr hole (0.5 mm diameter) was drilled on the skull. 1 μL of cell suspension (2 × 10^4^ cells/μL in PBS) was delivered at a depth of 3 mm over 2 min. Tumors were allowed to grow for seven days after implantation and then mini-osmotic pumps (Alzet^®^, DURECT, model 2004, Cupertino, CA, USA) were installed for local administration of drug to the tumor. Animals were anesthetized and a 3 mm brain infusion cannula connected to pump was set up at the previous tumor implantation site using brain infusion kit (Alzet^®^, DURECT). The pumps were placed subcutaneously on the mouse back. The drug was infused at 1 μL/h, 100 mg/mL of FA-CytC NPs over 14 days. Pumps with normal saline solution were used as a control.

Animals were inspected daily and body weight loss of 15%, decreased activity/responsiveness, abnormal posture or any neurological disorders signs were a subject for euthanasia. Time between tumor bearing and animal death was recorded. Comparison of survival curves were performed with use of log-rank (Mantel-Cox) test.

### 4.12. Statistical Analysis

Results are expressed as mean ± standard deviation (SD). Statistical probability was calculated using GraphPad software. Unpaired *t*-tests or one-way analysis of variance (ANOVA) tests followed by the Tukey’s post-hoc test were used to determine significance between groups. *p*-Values of less than 0.05 were considered significant. 

## Figures and Tables

**Figure 1 biomolecules-09-00154-f001:**
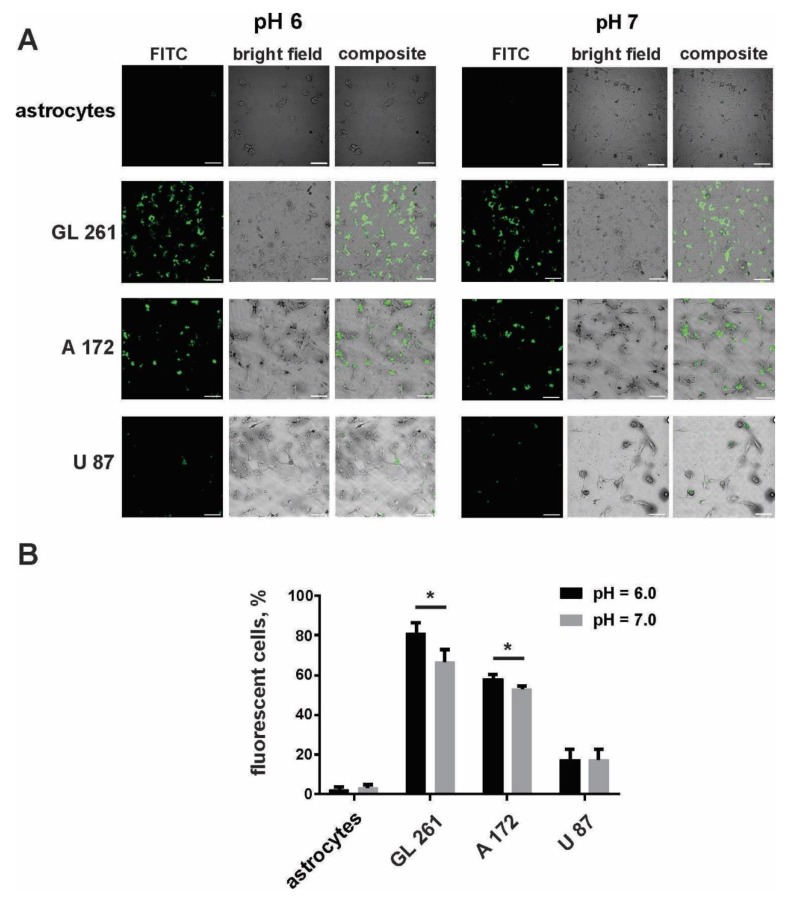
Some cultured glioma cells specifically take up FA-Cyt c NP constructs. Confocal images (**A**) and calculation (**B**) of live glioma cells and mouse primary cultured astrocytes were made after six hours of incubation with fluorescein isothiocyanate (FITC)-conjugated FA-Cyt c NPs at a concentration of 100 µg/mL. Mean ± S.E.; significant differences from control (*) are shown (*p* < 0.05). *N* = 3 (*N* is the number of the experimental repeats per cell culture). Scale bar, 50 µm.

**Figure 2 biomolecules-09-00154-f002:**
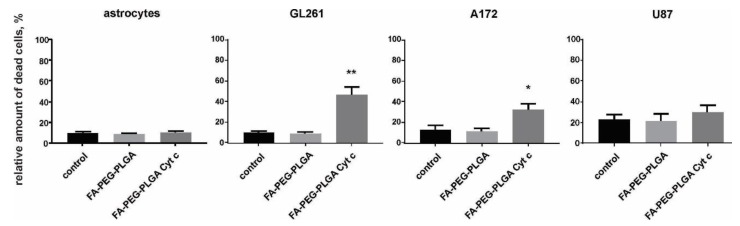
The viability of glioma cells and mouse primary cultured astrocytes treated with FA-coated Cyt c NPs (100 μg/mL). A live/dead assay based on calcein and ethidium homodimer-1 staining of live and dead cells was performed after 24 h of treatment with FA-Cyt c NPs. Quantitation of the number of dead cells as a percentage of the total number of cells is presented for the following treatments: control (untreated), FA-conjugated NPs not containing Cyt c (FA-PEG-PLGA), and FA-conjugated NPs containing Cyt c (FA-PEG- PLGA Cyt c). Mean ± S.E. and significant differences from control (* *p* < 0.05, ** *p* < 0.001) are shown. *N* = 5 (*N* is the number of the experimental repeats per cell culture).

**Figure 3 biomolecules-09-00154-f003:**
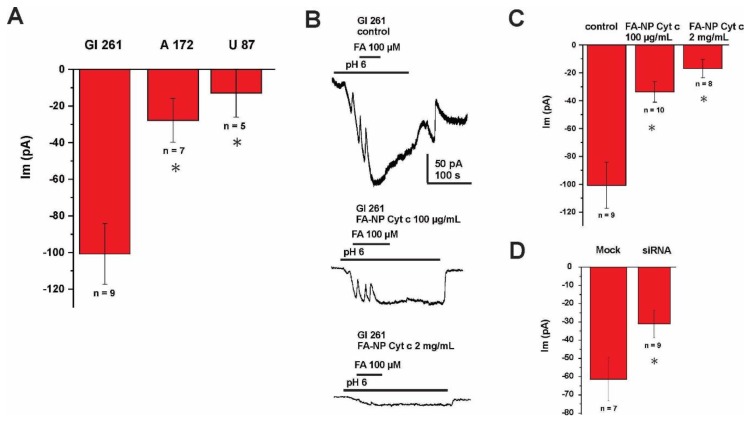
FA-induced membrane currents recorded from glioma cells. (**A**) Summary of currents recording from GL261, A172 and U87 glioma cells using whole-cell voltage clamp (held at −60 mV), elicited by addition of FA (100 µM) to the extracellular solution (pH 6.0). (**B**) Representative current recording and the summary of currents (**C**) from GL261 glioma cells using whole-cell voltage clamp (held at −60 mV), elicited by addition of FA (100 µM) to the extracellular solution (pH 6.0). The recording was performed with or without extracellular addition of FA-Cyt c NPs, given at different concentrations. (**D**) Summary of currents recorded from mock-transfected GL261 cells and cells transfected with siRNA against the PCFT, both induced by application of FA. Error bars represent SEM. * indicates a significant difference (*p* < 0.05). *n* is the number of cells investigated.

**Figure 4 biomolecules-09-00154-f004:**
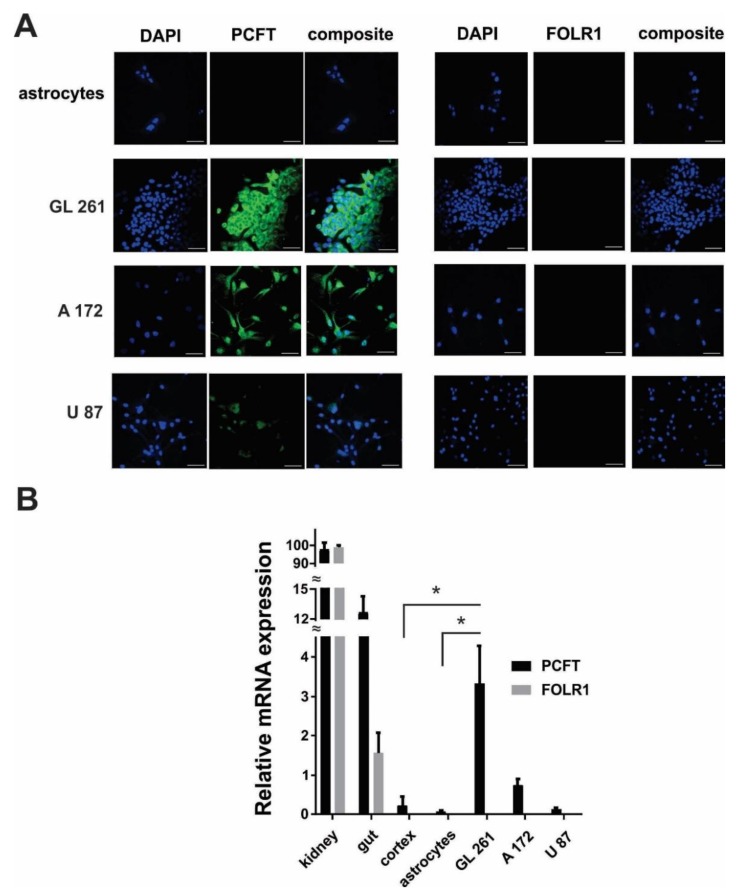
Proton-coupled folate transporter (PCFT) expression in GL261, A172, U87 glioma cells and mouse cultured primary astrocytes. (**A**) Confocal images showing immunofluorescent detection of the PCFT and folate receptor 1 (FOLR1) in glioma and astrocytic cells. DAPI (4′,6-diamidino-2-phenylindole) was used to stain the nuclei of cells. Scale bar, 50 µm. (**B**) Relative gene expression for PCFT and FOLR1 in astrocytic, U87, A172, GL261 cells, as well as in cortex, gut and kidney tissues in mice. Data are normalized to glyceraldehyde 3-phosphate dehydrogenase (GAPDH) as housekeeping gene. Mean ± SEM, significant differences from control (*) are shown (*p* < 0.001). *N* = 3 (*N* is the number of the experimental repeats per cell culture. In the case of tissue samples (kidney, gut, cortex) *N* is the number of tissue samples; each sample was analyzed in triplicate).

**Figure 5 biomolecules-09-00154-f005:**
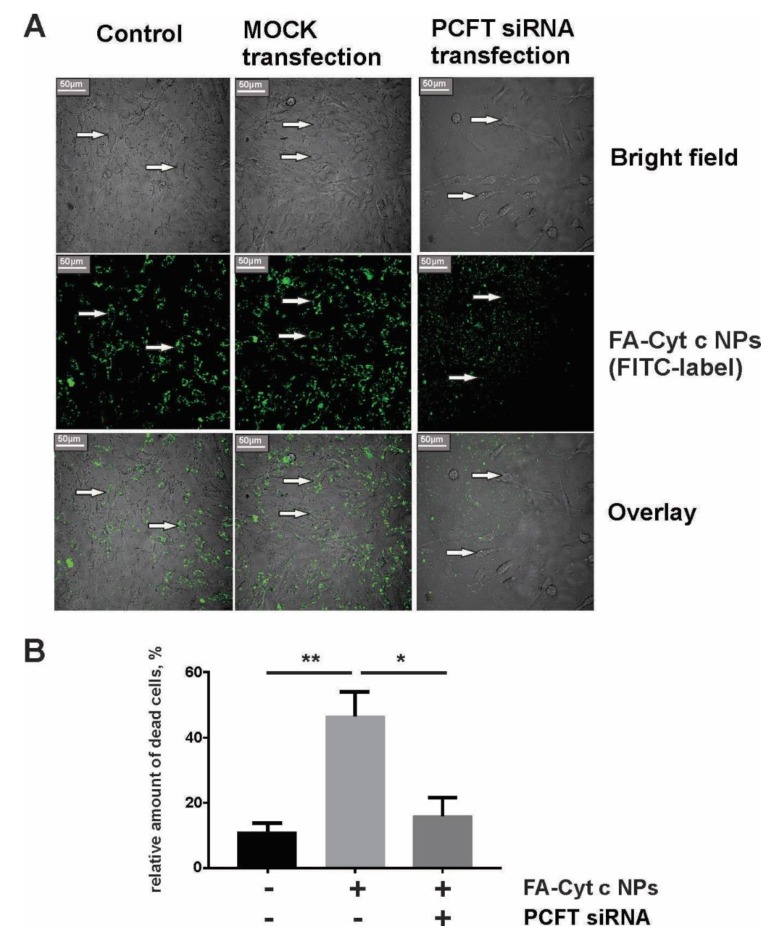
Internalization of FITC–FA-Cyt c NPs by GL261 cells transfected or untransfected with small interfering RNA (siRNA) against the PCFT. (**A**) Confocal imaging of live GL261 cells was performed after six hours of incubation with FITC-conjugated FA-Cyt c NPs. Control, MOCK-transfected (cells transfected without addition of siRNA) and PCFT siRNA-transfected cells are shown. Arrows indicate the localization of NPs in the cell cytoplasm. Scale bar, 50 µm. (**B**) A live/dead assay based on calcein and ethidium homodimer-1 staining of live and dead cells was performed for GL261 glioma cells, transfected or untransfected with siRNA against the PCFT, after 24 h of treatment with FA-Cyt c NPs (100 μg/mL). Quantitation of the number of dead cells as a percentage of the total number of cells is presented. Mean ± S.E. is shown. Asterisks ** and * indicate statistical significance at *p* < 0.001 and *p* < 0.05, respectively. *N* = 5 (*N* is the number of the experimental repeats per cell culture).

**Figure 6 biomolecules-09-00154-f006:**
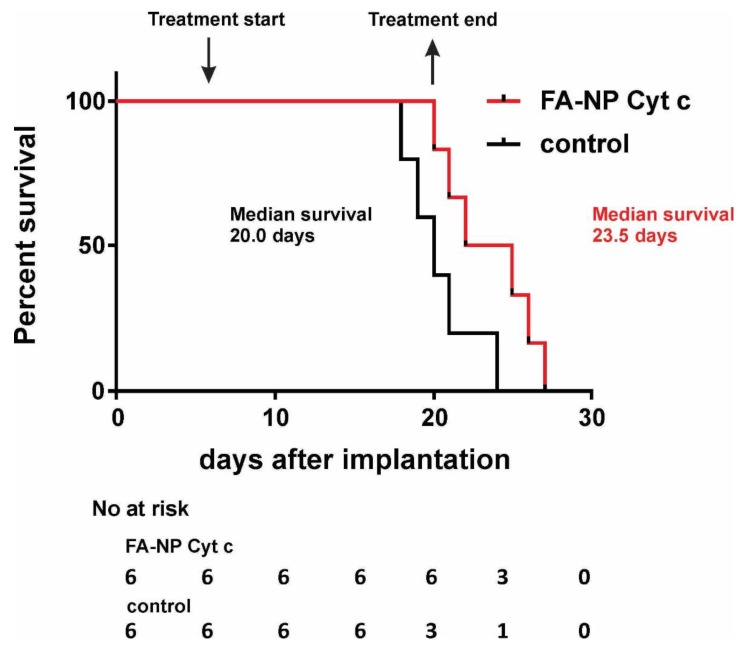
Local in-tumor infusion of FA-Cyt c NPs increases the survival rates in GL261/C57Bl/6 mouse glioma model. FA-Cyt c NPs (100 mg/mL diluted in PBS) were infused to the tumor at rate of 1 μL/h over 14 days, beginning on the 7th day after the tumor implantation, through mini-osmotic pumps. Control animals received PBS only. Kaplan-Meier survival analysis is shown. six animals per group were used. Curve comparison was performed with use of log-rank (Mantel-Cox) test (*p* < 0.05).

**Table 1 biomolecules-09-00154-t001:** Relative gene expression for proton-coupled folate transporter (PCFT) and folate receptor 1 (FOLR1) in human glioma cells purified from glioblastoma specimens. Data are normalized to glyceraldehyde 3-phosphate dehydrogenase (GAPDH) as housekeeping gene. Mean ± SEM and significant difference between PCFT and FOLR1 gene expression in the specimen (*p*) are shown. The PCFT data highlighted in bold have values that are significantly up-regulated compared to the FOLR1 expression in the specimen (*p* < 0.05). The color plot indicates the relative level of gene expression.

Patient Number	PCFT		FOLR1		*p*-Value
1	10.6 ± 1.51		84.13 ± 8.02		*p* ˂ 0.0001
2	**7.43 ± 0.60**		0.24 ± 0.15		*p* ˂ 0.0001
3	0.00		2626 ± 140.81		*p* ˂ 0.00001
4	**9.27 ± 0.75**		1.43 ± 0.25		*p* ˂ 0.0001
5	**5.5 ± 2.50**		1.03 ± 0.35		*p* ˂ 0.05
6	15.33 ± 1.53		38.33 ± 3.05		*p* ˂ 0.001
7	**1.03 ± 0.153**		0.5 ± 0.20		*p* ˂ 0.05
8	**10.5 ± 1.50**		0.27 ± 0.13		*p*˂ 0.001
9	**379.67 ± 35.92**		109.33 ± 17.78		*p* ˂ 0.001

0.00–1.00
1.00–3.00
3.00–9.00
9.00–25.00
25.00–200.00
>200

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
