# Peer review of "Targeted Delivery of Nanoparticulate Cytochrome C into Glioma Cells Through the Proton-Coupled Folate Transporter"

_biomolecules, 2019, doi:10.3390/biom9040154_

Reviewer 1 Report

In this study the authors preseted proton‐coupled folate transporter (PCFT) as a possible route for targeted drug delivery to some gliomas. The results are convincing and well documented. 

Comments and questions:

Is there any study on the investigation of expression and function of PCFT in neurons, pericytes and brain capillary endothelial cells? If there is, please add it to the discussion. The authors described, that the cancer cells are highly dependent on FA for DNA synthesis and replication, and the overexpression of FA receptors demonstrated for many cancers compared with noncancerous cells. In this case why is it important to deliver the drug by osmotic minipump ditrectly into the gliomas? It would be suggested to use a less invasive drug administration method like intranasal delivery. Please add a citation to your explanation (Evaluation of intranasal delivery route of drug administration for brain targeting. Erdő F, Bors LA, Farkas D, Bajza Á, Gizurarson S.Brain Res Bull. 2018 Oct;143:155-170). What kind of side effect of the FA‐coated Cyt c NPs could be expected if the drug is given intranasally?

Otherwise the manuscript includes very interesting and important new results. After a minor revision I suggest it for publication.

Author Response

We thank the reviewer for the suggestions that helped to improve the paper and to consider another method for NPs delivery in future studies. Below we provide point by point response:

1.       Is there any study on the investigation of expression and function of PCFT in neurons, pericytes and brain capillary endothelial cells? If there is, please add it to the discussion.

It was shown that the PCFT expressed in choroid plexus epithelium. We highlighted it in the Discussion section.

2.       The authors described, that the cancer cells are highly dependent on FA for DNA synthesis and replication, and the overexpression of FA receptors demonstrated for many cancers compared with noncancerous cells. In this case why is it important to deliver the drug by osmotic minipump ditrectly into the gliomas? It would be suggested to use a less invasive drug administration method like intranasal delivery. Please add a citation to your explanation (Evaluation of intranasal delivery route of drug administration for brain targeting. Erdő F, Bors LA, Farkas D, Bajza Á, Gizurarson S.Brain Res Bull. 2018 Oct;143:155-170). What kind of side effect of the FA‐coated Cyt c NPs could be expected if the drug is given intranasally?

This discussion was added in lines 339-349.

Reviewer 2 Report

In the manuscript by Kucheryavykh et al. the authors assessed the possibility to target glioma cells using folic-acid conjugated Cytochrome C particles. They show that these NPs can be used to target a subset of cells with good to very good rates and assessed the receptor of these NPs. Finally they analyzed some human samples and analyzed a previously published in vivo experiment showing moderate increases in survival. The identification of the molecular receptor is a key step towards a targeted therapy and therefore this study can be considered as important.

The manuscript is overall well written, although some typos can still be found, while other sentences could be edited for brevity. Prior publication some weaknesses should be addressed:

Major points:

1.       In their introduction the authors state that FA-Cyt C NPs are a novel development of previously used Cyt C NPs which they used for glioma treatment in vitro and in vivo.  Apparantly, the addition of FA to these NPs should improve its delivery. To consolidate this claim the other should prove that FA Cyt C NPs are indeed superior to unlabeled NPs. This should be shown in terms of delivery and (if possible) also for cell death induction.

2.       The effects on cell death might be considered as minor. The author provide data after 24h of treatment, but also show that an extended treatment of astrocytes (up to 5 days) does not result in unspecific cell deat. What about induction of cell death in the “sensitive” lines? Can it be increased by extended treatment? Please comment or show.

3.       The labels for the NPs is different in each figure and figure legend (e.g. FA-PEG-PLGA Cyt C NP in Fig.2 and FA-NP Cyt C in Fig. 3). Are these similar particles or are they chemically distinct. If they are similar, why do the authors name them differently?

4.       The authors identify the PCFT as the likely target of their NPs. Does PCFT-Knockdown also protect from cell death induction? And accordingly, does PCFT-overexpression (e.g. in U87) sensitize the cells towards cell death induction?

5.       The authors use their NPs in vivo in a dosage 1,000 fold higher than in vitro (100 mg/ml vs. 100 µg/ml). These differences seem quite considerate. Please comment.

6.       The authors used Cytochrome C to induce cell death using their NPs. Does the cellular response correlate with the sensitivity towards known inducer of apoptosis (e.g. TMZ, ABT737)? Please discuss.

7.       This system can potentially be used to deliver other therapeutics into tumor cells. What are the limitations (cargo size, polarity, hydrophobicity, etc.)? Please discuss.

8.       The discussion starts with the sentence implying that both PCFT and FOLR1 are required for NP transfection. However, the authors showed very low expression of FOLR1 in the analyzed cell lines (Fig. 4). Why should FOLR1 be important? Please further discuss.

Minor points:

1.       Fig. 1: The authors report the relative amount of fluorescent cells. The axis is unnecessary long, making it difficult to read. An axis-title like “fluorescent cells [%], might be sufficient. The cells in pH7 (especially A172 and U87) look different than in pH6. Please assess cell viability and comment on the toxicity.

2.       Fig. 2: 20% cell death in U87 of untreated cells is quite high. Please comment.

3.       Fig. 4: The picture are quite small. The authors should place the bar graphs below the micrographs and increase their size to improve readability.

4.       Fig. 5: It appears as if mock-transfection increases the amount of FITC-positive cells. Please quantify the amount of FITC-positive cells.

5.       The authors routinely write N=X. What does N mean exactly? Number of experiments or number of samples? If experiments à how many samples were analyzed; if samples à how often were the experiments repeated?

6.       The authors analyzed data from the TCGA database (lines 202-208) but do not show their analyses. Please include those and specify which dataset was analyzed.

7.       Table 1: The box widths are different. These should be equalized. Also, the color coding should be shown adjacent to each other to make the comparisons easier.

8.       Fig. 6: How many animals were used (per group) this should written in the main text and in MatMet. Treatment start should be marked in the figure. The result of the statistical analysis should also be given in the figure.

9.       The authors assessed CNV for PCFT. This seems a rather specific analysis. Please elaborate why CNV was chosen. Also, more general analyses should conducted (e.g. expression in various glioma grades; prognostic value etc.). This can easily be done using various freely available online tools and would certainly improve the manuscripts overall significance.

10.   Fig. S2: The authors state N=3, but no standard deviation is shown. Please re-check.

11.   The authors only show p<0.05 denoted by “*”. The authors should provide more detail by adding “**”, “***” and “****” where applicable.

12.   Line 192: space between “in” and “vestigated”.

13.   Line 217: the past tense of to lead is “lead”

14.   Line 224: This sentence should be re-written.

15.   Line 337: Bracket around the citation are missing

16.   Line 405: 4 in “104” is not in upper case. Also 5.0 might be shortened to 5

17.   Line 448: same mistake as line 405

Author Response

We thank the reviewer for the constructive suggestions that helped to improve the paper. Below we provide point by point response:

Major points:

1.            In their introduction the authors state that FA-Cyt C NPs are a novel development of previously used Cyt C NPs which they used for glioma treatment in vitro and in vivo.  Apparently, the addition of FA to these NPs should improve its delivery. To consolidate this claim the other should prove that FA Cyt C NPs are indeed superior to unlabeled NPs. This should be shown in terms of delivery and (if possible) also for cell death induction.

The Cyt c nano-constructs were fully characterized in our previous publication (Morales-Cruz, M, Cruz-Montañez, A, Figueroa, CM, González-Robles, T, Davila, J, Inyushin, M, et al. Combining Stimulus-Triggered Release and Active Targeting Strategies Improves Cytotoxicity of Cytochrome c Nanoparticles in Tumor Cells. Mol Pharm., 2016. 13 (8): 2844–2854. doi: 10.1021/acs.molpharmaceut.6b00461). Below we provide the Figures taken from this publication where we demonstrate the efficacy of folate-decorated Cyt c-based NPs compared to folate-free Cyt c-based NPs. Immunocytochemistry and cell viability approaches were used. Enhanced cytotoxic effect of FA-PEG-PLGA-coated Cyt c-based NPs was shown with use of propidium Iodide stating (Fig 5A). Reduction of cell viability for Cyt c NPs coated with the FA was 80% compared to 20% for non-coated NPs (Fig 5B). Endosomal entrapment of FA targeted Cyt c NPs is shown in Fig 5C and confirms the uptake by endocytosis.

The effect of folate-decorated NPs on folate-deficient cell lines was tested in Fig 6 of the mentioned published paper (provided below).

We added clarifications and a reference to this study in lines 78-80 of re-submitted manuscript.

2.       The effects on cell death might be considered as minor. The authors provide data after 24h of treatment, but also show that an extended treatment of astrocytes (up to 5 days) does not result in unspecific cell death. What about induction of cell death in the “sensitive” lines? Can it be increased by extended treatment? Please comment or show.

Effect of Cyt c NPs on “sensitive” cells was addressed in the comment 1. The activated Akt signaling in gliomas can provide the possible explanation for the lower sensitivity to Cyt c, compared to mesenchymal tumors (as HeLa cell line).  Currently we are in preparation of a research article where we demonstrate that PI3K/Akt blockers sensitize glioma cells to Cyt c NPs and reduce cell viability till 10% of the total amount of cells during 12 hours of treatment. We added some speculations about the possible role of RTK/Akt axis in glioma cell resistance to Cyt c in lines 321-331 of the current re-submitted paper, however, the complete study will be a subject of separate publication.

3.       The labels for the NPs is different in each figure and figure legend (e.g. FA-PEG-PLGA Cyt C NP in Fig.2 and FA-NP Cyt C in Fig. 3). Are these similar particles or are they chemically distinct. If they are similar, why do the authors name them differently?

Fig. 2 demonstrates the cytotoxic effect of a complete construct (FA-PEG-PLGA-NP Cyt c) and of a part of the construct not containing Cyt c (FA-PEG-PLGA). We keep this labeling in Fig. 2 in order to identify complete NPs containing Cyt c and FA-PEG-PLGA polymer. We removed “NP” from the construct identification in the figure and figure legend in order to avoid confusion with other figures.

4.       The authors identify the PCFT as the likely target of their NPs. Does PCFT-Knockdown also protect from cell death induction? And accordingly, does PCFT-overexpression (e.g. in U87) sensitize the cells towards cell death induction?

We added new data (Fig. 5B) indicating that PCFT knockdown reduces the sensitivity of GL261 glioma cells to FA-Cyt c NP. Description of data is added in lines 194-198.

5.       The authors use their NPs in vivo in a dosage 1,000 fold higher than in vitro (100 mg/ml vs. 100 µg/ml). These differences seem quite considerate. Please comment.

NPs stock solution concentration was used in mini-osmotic pumps. Taking in account the rate of in-tissue delivery (1 μL/h), diffusion distance of NPs in tumor tissue (1500 μm/24 hours, determined in in vivo studies), internalization by tumor cells and in-tissue degradation, we consider that in vivo concentration of 100mg/ml is equivalent to in vitro concentration of 100 µg/ml. Additionally, 100 mg/ml was determined as the most effective concentration in in vivo dose response studies, referring to our previous publications.

6.       The authors used Cytochrome C to induce cell death using their NPs. Does the cellular response correlate with the sensitivity towards known inducer of apoptosis (e.g. TMZ, ABT737)? Please discuss.

Added to the Discussion section, lines 325-331.

7.       This system can potentially be used to deliver other therapeutics into tumor cells. What are the limitations (cargo size, polarity, hydrophobicity, etc.)? Please discuss.

Added to the Discussion section, lines 332-338.

8.       The discussion starts with the sentence implying that both PCFT and FOLR1 are required for NP transfection. However, the authors showed very low expression of FOLR1 in the analyzed cell lines (Fig. 4). Why should FOLR1 be important? Please further discuss.

It was a wording inaccuracy in the mentioned sentence that made an impression that both FOLR1 and PCFT are needed for NPs internalization. We corrected the sentence in order to make it clear, that either FOLR1 or PCFT is needed. Previously the FA carriers were used mostly for cancers expressing FOLR1. We demonstrated that cancers expressing PCFT also can be targeted by FA-based carriers.

Minor points:

1.       Fig. 1: The authors report the relative amount of fluorescent cells. The axis is unnecessary long, making it difficult to read. An axis-title like “fluorescent cells [%], might be sufficient. The cells in pH7 (especially A172 and U87) look different than in pH6. Please assess cell viability and comment on the toxicity.

Fig 1 is fixed. The cell viability is provided in Supplementary Figure S1.

2.       Fig. 2: 20% cell death in U87 of untreated cells is quite high. Please comment.

The standard amount of dead cells in U87 culture is 15-20% if the confluence is above 60%. This is slightly higher compared to other glioma cell lines and probably related to specific mutations in DNA reparation mechanisms in these cells.

3.       Fig. 4: The pictures are quite small. The authors should place the bar graphs below the micrographs and increase their size to improve readability.

The Figure 4 is fixed.

4.       Fig. 5: It appears as if mock-transfection increases the amount of FITC-positive cells. Please quantify the amount of FITC-positive cells.

The viability of transfected cells are now provided in Supplementary Figure S4. Description of data is added in lines 194-198.

5.       The authors routinely write N=X. What does N mean exactly? Number of experiments or number of samples? If experiments à how many samples were analyzed; if samples à how often were the experiments repeated?

Identification of N is now provided for each figure legend.

6.       The authors analyzed data from the TCGA database (lines 202-208) but do not show their analyses. Please include those and specify which dataset was analyzed.

We obtained statistical data provided at the TCGA for PCFT and FOLR1 expression in GBMs, but we did not perform analysis of specific datasets. We state this in Methodology section.

7.       Table 1: The box widths are different. These should be equalized. Also, the color coding should be shown adjacent to each other to make the comparisons easier.

The table is fixed.

8.       Fig. 6: How many animals were used (per group) this should written in the main text and in MatMet. Treatment start should be marked in the figure. The result of the statistical analysis should also be given in the figure.

The figure 6 is fixed.

9.       The authors assessed CNV for PCFT. This seems a rather specific analysis. Please elaborate why CNV was chosen. Also, more general analyses should conducted (e.g. expression in various glioma grades; prognostic value etc.). This can easily be done using various freely available online tools and would certainly improve the manuscripts overall significance.

Statistical data for CNV in GBM were obtained from TCGA because the level of protein expression of PCFT in gliomas directly depends on this parameter due to aneuploidy of cancer cells and related gain or loss of the gene.

We added statistical information related to low grade glioma in lines 225-228. We did not add the prognostic and survival information related to the expression of PCFT and FOLR1 in glioma patient in order to keep the paper focused on justification of use of FA-NPs in gliomas. Investigation of folate carrier’s expression in relation to tumor behavior is not in scope of this study. 

10.   Fig. S2: The authors state N=3, but no standard deviation is shown. Please re-check.

Standard deviation is shown but it has small amplitude on this graph.

11.   The authors only show p<0.05 denoted by “*”. The authors should provide more detail by adding “**”, “***” and “****” where applicable.

This was fixed where applicable.

12.   Line 192: space between “in” and “vestigated”.

fixed

13.   Line 217: the past tense of to lead is “lead”

fixed

14.   Line 224: This sentence should be re-written.

fixed

15.   Line 337: Bracket around the citation are missing

fixed

16.   Line 405: 4 in “104” is not in upper case. Also 5.0 might be shortened to 5

fixed

17.   Line 448: same mistake as line 405

fixed

Round  2

Reviewer 2 Report

Author's Notes

The authors have considerably improved the manuscript and the open questions were well to very good answered.

We thank the reviewer for the constructive suggestions that helped to improve the paper. Below we provide point by point response:

Major points:

1.            In their introduction the authors state that FA-Cyt C NPs are a novel development of previously used Cyt C NPs which they used for glioma treatment in vitro and in vivo.  Apparently, the addition of FA to these NPs should improve its delivery. To consolidate this claim the other should prove that FA Cyt C NPs are indeed superior to unlabeled NPs. This should be shown in terms of delivery and (if possible) also for cell death induction.

The Cyt c nano-constructs were fully characterized in our previous publication (Morales-Cruz, M, Cruz-Montañez, A, Figueroa, CM, González-Robles, T, Davila, J, Inyushin, M, et al. Combining Stimulus-Triggered Release and Active Targeting Strategies Improves Cytotoxicity of Cytochrome c Nanoparticles in Tumor Cells. Mol Pharm., 2016. 13 (8): 2844–2854. doi: 10.1021/acs.molpharmaceut.6b00461). Below we provide the Figures taken from this publication where we demonstrate the efficacy of folate-decorated Cyt c-based NPs compared to folate-free Cyt c-based NPs. Immunocytochemistry and cell viability approaches were used. Enhanced cytotoxic effect of FA-PEG-PLGA-coated Cyt c-based NPs was shown with use of propidium Iodide stating (Fig 5A). Reduction of cell viability for Cyt c NPs coated with the FA was 80% compared to 20% for non-coated NPs (Fig 5B). Endosomal entrapment of FA targeted Cyt c NPs is shown in Fig 5C and confirms the uptake by endocytosis.

The effect of folate-decorated NPs on folate-deficient cell lines was tested in Fig 6 of the mentioned published paper (provided below).

We added clarifications and a reference to this study in lines 78-80 of re-submitted manuscript.

agreed

2.       The effects on cell death might be considered as minor. The authors provide data after 24h of treatment, but also show that an extended treatment of astrocytes (up to 5 days) does not result in unspecific cell death. What about induction of cell death in the “sensitive” lines? Can it be increased by extended treatment? Please comment or show.

Effect of Cyt c NPs on “sensitive” cells was addressed in the comment 1. The activated Akt signaling in gliomas can provide the possible explanation for the lower sensitivity to Cyt c, compared to mesenchymal tumors (as HeLa cell line).  Currently we are in preparation of a research article where we demonstrate that PI3K/Akt blockers sensitize glioma cells to Cyt c NPs and reduce cell viability till 10% of the total amount of cells during 12 hours of treatment. We added some speculations about the possible role of RTK/Akt axis in glioma cell resistance to Cyt c in lines 321-331 of the current re-submitted paper, however, the complete study will be a subject of separate publication.

agreed

3.       The labels for the NPs is different in each figure and figure legend (e.g. FA-PEG-PLGA Cyt C NP in Fig.2 and FA-NP Cyt C in Fig. 3). Are these similar particles or are they chemically distinct. If they are similar, why do the authors name them differently?

Fig. 2 demonstrates the cytotoxic effect of a complete construct (FA-PEG-PLGA-NP Cyt c) and of a part of the construct not containing Cyt c (FA-PEG-PLGA). We keep this labeling in Fig. 2 in order to identify complete NPs containing Cyt c and FA-PEG-PLGA polymer. We removed “NP” from the construct identification in the figure and figure legend in order to avoid confusion with other figures.

agreed

4.       The authors identify the PCFT as the likely target of their NPs. Does PCFT-Knockdown also protect from cell death induction? And accordingly, does PCFT-overexpression (e.g. in U87) sensitize the cells towards cell death induction?

We added new data (Fig. 5B) indicating that PCFT knockdown reduces the sensitivity of GL261 glioma cells to FA-Cyt c NP. Description of data is added in lines 194-198.

agreed. Maybe the overexpression might be analyzed in future studies, since it would likely open this approach up to more cell types. But this is just an interested suggestion and no criticism.

5.       The authors use their NPs in vivo in a dosage 1,000 fold higher than in vitro (100 mg/ml vs. 100 µg/ml). These differences seem quite considerate. Please comment.

NPs stock solution concentration was used in mini-osmotic pumps. Taking in account the rate of in-tissue delivery (1 μL/h), diffusion distance of NPs in tumor tissue (1500 μm/24 hours, determined in in vivo studies), internalization by tumor cells and in-tissue degradation, we consider that in vivo concentration of 100mg/ml is equivalent to in vitro concentration of 100 µg/ml. Additionally, 100 mg/ml was determined as the most effective concentration in in vivo dose response studies, referring to our previous publications.

agreed.

6.       The authors used Cytochrome C to induce cell death using their NPs. Does the cellular response correlate with the sensitivity towards known inducer of apoptosis (e.g. TMZ, ABT737)? Please discuss.

Added to the Discussion section, lines 325-331.

My question was rather aimed at whether cells that are insensitive toward small molecule-mediated apoptosis-induction are also insensitive towards Cyt C delivery, but the added discussion is well enough, so agreed.

7.       This system can potentially be used to deliver other therapeutics into tumor cells. What are the limitations (cargo size, polarity, hydrophobicity, etc.)? Please discuss.

Added to the Discussion section, lines 332-338.

In combination with the added discussion in lines 325-330 this is acceptable.

8.       The discussion starts with the sentence implying that both PCFT and FOLR1 are required for NP transfection. However, the authors showed very low expression of FOLR1 in the analyzed cell lines (Fig. 4). Why should FOLR1 be important? Please further discuss.

It was a wording inaccuracy in the mentioned sentence that made an impression that both FOLR1 and PCFT are needed for NPs internalization. We corrected the sentence in order to make it clear, that either FOLR1 or PCFT is needed. Previously the FA carriers were used mostly for cancers expressing FOLR1. We demonstrated that cancers expressing PCFT also can be targeted by FA-based carriers.

agreed.

Minor points:

1.       Fig. 1: The authors report the relative amount of fluorescent cells. The axis is unnecessary long, making it difficult to read. An axis-title like “fluorescent cells [%], might be sufficient. The cells in pH7 (especially A172 and U87) look different than in pH6. Please assess cell viability and comment on the toxicity.

Fig 1 is fixed. The cell viability is provided in Supplementary Figure S1.

2.       Fig. 2: 20% cell death in U87 of untreated cells is quite high. Please comment.

The standard amount of dead cells in U87 culture is 15-20% if the confluence is above 60%. This is slightly higher compared to other glioma cell lines and probably related to specific mutations in DNA reparation mechanisms in these cells.

3.       Fig. 4: The pictures are quite small. The authors should place the bar graphs below the micrographs and increase their size to improve readability.

The Figure 4 is fixed.

4.       Fig. 5: It appears as if mock-transfection increases the amount of FITC-positive cells. Please quantify the amount of FITC-positive cells.

The viability of transfected cells are now provided in Supplementary Figure S4. Description of data is added in lines 194-198.

5.       The authors routinely write N=X. What does N mean exactly? Number of experiments or number of samples? If experiments à how many samples were analyzed; if samples à how often were the experiments repeated?

Identification of N is now provided for each figure legend.

6.       The authors analyzed data from the TCGA database (lines 202-208) but do not show their analyses. Please include those and specify which dataset was analyzed.

We obtained statistical data provided at the TCGA for PCFT and FOLR1 expression in GBMs, but we did not perform analysis of specific datasets. We state this in Methodology section.

7.       Table 1: The box widths are different. These should be equalized. Also, the color coding should be shown adjacent to each other to make the comparisons easier.

The table is fixed.

the color now makes much more sense.

8.       Fig. 6: How many animals were used (per group) this should written in the main text and in MatMet. Treatment start should be marked in the figure. The result of the statistical analysis should also be given in the figure.

The figure 6 is fixed.

Minor point: There are two labels "Treatment start". I assume the right one should read "treatment end"?

9.       The authors assessed CNV for PCFT. This seems a rather specific analysis. Please elaborate why CNV was chosen. Also, more general analyses should conducted (e.g. expression in various glioma grades; prognostic value etc.). This can easily be done using various freely available online tools and would certainly improve the manuscripts overall significance.

Statistical data for CNV in GBM were obtained from TCGA because the level of protein expression of PCFT in gliomas directly depends on this parameter due to aneuploidy of cancer cells and related gain or loss of the gene.

We added statistical information related to low grade glioma in lines 225-228. We did not add the prognostic and survival information related to the expression of PCFT and FOLR1 in glioma patient in order to keep the paper focused on justification of use of FA-NPs in gliomas. Investigation of folate carrier’s expression in relation to tumor behavior is not in scope of this study. 

10.   Fig. S2: The authors state N=3, but no standard deviation is shown. Please re-check.

Standard deviation is shown but it has small amplitude on this graph.

black deviation on black boxes are hard to read. Please consider a different color scheme in this case.

11.   The authors only show p<0.05 denoted by “*”. The authors should provide more detail by adding “**”, “***” and “****” where applicable.

This was fixed where applicable.

12.   Line 192: space between “in” and “vestigated”.

fixed

13.   Line 217: the past tense of to lead is “lead”

fixed

14.   Line 224: This sentence should be re-written.

fixed

15.   Line 337: Bracket around the citation are missing

fixed

16.   Line 405: 4 in “104” is not in upper case. Also 5.0 might be shortened to 5

fixed

17.   Line 448: same mistake as line 405

fixed